# Understanding Healthcare Professionals’ Perspectives on Point-of-Care Testing

**DOI:** 10.3390/diagnostics12020533

**Published:** 2022-02-19

**Authors:** Sean Teebagy, Ziyue Wang, Denise Dunlap, Connor Saleeba, Danielle DiMezza, JoAnn Crain, Craig M. Lilly, Bryan Buchholz, David D. McManus, Nathaniel Hafer

**Affiliations:** 1Department of Medicine, UMass Chan Medical School, 55 Lake Ave. N, Worcester, MA 01655, USA; sean.teebagy@umassmed.edu (S.T.); ziyue.wang@umassmed.edu (Z.W.); connor.saleeba@umassmed.edu (C.S.); danielle.dimezza@aol.com (D.D.); craig.lilly@umassmed.edu (C.M.L.); david.mcmanus@umassmed.edu (D.D.M.); 2Manning School of Business, UMass Lowell, 220 Pawtucket St., Lowell, MA 01854, USA; denise_dunlap@uml.edu; 3Clinical Laboratory Operations, UMass Memorial Health, 55 Lake Ave. N, Worcester, MA 01655, USA; joann.crain@umassmemorial.org; 4Department of Anesthesiology and Perioperative Medicine, UMass Chan Medical School, 55 Lake Ave. N, Worcester, MA 01655, USA; 5Department of Surgery, UMass Chan Medical School, 55 Lake Ave. N, Worcester, MA 01655, USA; 6Department of Biomedical Engineering, UMass Lowell, 220 Pawtucket St., Lowell, MA 01854, USA; bryan_buchholz@uml.edu; 7Department of Population and Quantitative Health Sciences, UMass Chan Medical School, 55 Lake Ave. N, Worcester, MA 01655, USA; 8UMass Center for Clinical and Translational Science, UMass Chan Medical School, 55 Lake Ave. N, Worcester, MA 01655, USA; 9Program in Molecular Medicine, UMass Chan Medical School, 55 Lake Ave. N, Worcester, MA 01655, USA

**Keywords:** point-of-care testing, rapid testing, healthcare provider survey

## Abstract

Point-of-care testing (POCT) is an emerging technology that provides crucial assistance in delivering healthcare. The COVID-19 pandemic led to the accelerated importance of POCT technology due to its in-home accessibility. While POCT use and implementation has increased, little research has been published about how healthcare professionals perceive these technologies. The objective of our study was to examine the current perspectives of healthcare professionals towards POCT. We surveyed healthcare professionals to quantify perceptions of POCT usage, adoption, benefits, and concerns between October 2020 and November 2020. Questions regarding POCT perception were assessed on a 5-point Likert Scale. We received a total of 287 survey responses. Of the respondents, 53.7% were male, 66.6% were white, and 30.7% have been in practice for over 20 years. We found that the most supported benefit was POCTs ability to improve patient management (92%) and that the most supported concern was that POCTs lead to over-testing (30%). This study provides a better understanding of healthcare workers’ perspectives on POCT. To improve patient outcomes through the usage of POCT, greater research is needed to assess the needs and concerns of industry and healthcare stakeholders.

## 1. Introduction

The COVID-19 pandemic led to the rapid implementation of society-wide non-pharmaceutical healthcare interventions, such as lockdowns and telemedicine. This shift, combined with other state and federal regulations, created a unique interest in and demand for point-of-care testing (POCT) solutions [1,2]. The COVID-19 pandemic demonstrated the necessity of utilizing and understanding best use case scenarios for POCT. Furthermore, prior to the pandemic, the advantages of expanding the role of patients in the management of their own health and wellness through the use of POCTs were widely recognized. For example, an increasing number of Americans are caring for aging community members in settings ideal for POCT solutions [3,4]. Synchronous changes in population health, along with public health mandates, have created a unique environment and lead to an unprecedented opportunity for more efficient and patient-centered care using POCT. POCT products have the potential to both expedite diagnosis and support patient-integrated management, while also reducing transportation cost, delays, and caregiver inconvenience [5,6]. While prior work has documented the advantages of POCT, little has been published about the adoption of POCTs by healthcare providers. 

It is essential to understand the perceived utility and adoption of this technology by various healthcare professionals in order to increase the successful implementation and use of POCTs by clinicians. We build upon a recent study that surveyed healthcare professionals to ascertain their opinions on POCTs in 2019, which specifically compared responses from those in cardiovascular medicine to other healthcare professionals [7]. Here, we discuss the results of a similar survey sent to healthcare professionals in November 2020 to further build the knowledge base of healthcare providers’ perspectives of POCT. We found that most respondents perceived an unmet need for POCTs, a majority of participants agreed with statements related to the benefits to POCT, and a minority agreed with statements describing the possible concerns over POCT.

## 2. Materials and Methods

A diverse group of participants with expertise in healthcare was invited to participate in this POCT survey. The group was made up of mostly healthcare professionals, healthcare researchers, and developers. Survey invitations were distributed via email to all potential respondents via 16 internal and external email directories. The total number of individuals reached via these directories is unknown, but is greater than 15,000. Certain external email directories were chosen with the goal of inviting a diverse group of healthcare professionals interested in POCT. Internal email directories were originally established in 2019, when we conducted a similar survey and distribution strategy. These individuals were identified via the National Institutes of Health Reporter (https://projectreporter.nih.gov/reporter.cfm, 24 October 2020), Profiles (https://profiles.umassmed.edu/search/, 22 October 2020), and Direct2experts (http://direct2experts.org/, 22 October 2020) [7]. Furthermore, we recruited participants via a LinkedIn invitation post. More details on the mailing list used can be found in Figure 1. The only exclusion criteria was if someone did not self-identify as a healthcare worker. The survey was launched on 29 October 2020 and closed on 30 November 2020. If potential survey respondents did not reply, a second reminder email was sent a few weeks following the initial invitation. A total of 287 respondents completed the survey. This study was deemed to be exempt from review by the Institutional Review Board (IRB) in July 2019 by the UMass Chan Medical School’s IRB (docket # H00018195).

### 2.1. Survey, Data Collection and Storage

The survey is based on a previously validated questionnaire that was used in our 2019 survey, which specifically compared responses from those in cardiovascular medicine to other healthcare professionals (see Appendix A). In 2019, an internal team of both clinical medicine and business development experts participated in developing this survey as described previously [7]. In the development of our 2020 survey, questions were added regarding the impact of COVID-19. In many cases, including questions regarding the adoption of POCT, the questions listed in this survey were the same questions asked in the 2019 survey.

In accordance with our similar survey on POCT from 2019, this survey contained five elements. The first element covered demographics, including gender, profession, patient practice environment, and years in practice. The second element contained open text fields, which allowed respondents to list diseases that could benefit from POCT. The third element covered respondents’ perceived benefits of and concerns over POCT. The fourth element covered the product adoption practices of POCT technologies. Finally, the fifth element covered POCT use during the ongoing COVID-19 pandemic.

Similar to our 2019 survey, this survey contained questions that measured general POCT usage, benefits, and concerns. Additionally, the business-related aspects of POCT technology use and adoption were included to better understand the perspectives of individuals in the healthcare industry. The questions measuring general POCT matters were adapted from the National Heart, Lung, and Blood Institute (NHLBI) strategic vision published in 2016 [8], and from a survey developed by researchers from the Point-of-Care Technology Research Network (POCTRN) center located at Johns Hopkins University [9]. Questions regarding the business-related aspects of healthcare technology were adapted from two seminal studies focused on the adoption of new technologies [10,11].

Analogous to our 2019 survey, most questions were assessed on a Likert-like scale, allowing participants to select “strongly disagree,” “disagree,” “neutral,” “agree,” or “strongly agree.” Demographics information was collected via multiple-choice questions or through open-ended text boxes. Participants were further asked to list up to five conditions for which POCT could help with: (1) diagnosis of a disease, and (2) management or monitoring of a disease. Participants’ answers to these two questions were through open-ended text boxes.

The survey was generated by a REDCap generated interface. All data were received from participants and transmitted directly into the study server for storage. The server is hosted by the UMass Chan network and was only accessed by authorized individuals.

### 2.2. Data Analysis

Open-ended responses regarding medical specialties were categorized using an adapted list of standard medical specialties [12]. The variables from questions that allow participants to answer on the 5-point Likert-like scale described above were collected into two categories: (1) responses indicated “strongly agree” and “agree” were categorized into agreement, and (2) “strongly disagree” and “disagree” were designated as disagreement. Any “neutral” response was excluded from the analysis. Analysis of all the data from survey respondents was limited to frequency calculations. Data analysis was completed in SAS version 9.3. Counts of point-of-care glucose tests were retrieved from UMass Memorial Health records dating from 2018–2021.

## 3. Results

### 3.1. Demographics

Participants were asked about their demographic information in order to evaluate if a large group of caregivers from a broad spectrum of specialties and backgrounds were surveyed about POCT. Furthermore, continued collection of demographic information will allow analysis of POCT perspective based on location. A total of 287 participants responded to the survey. Of those, 154 (53.7%) were male, 120 (41.8%) were female, and 13 (4.5%) were other. In regards to race, 191 (67.3%) were white, 9 (3.2%) were black, 47 (16.5%) were Asian, 1 (0.4%) was Alaskan Native or Native American, 4 (1.4%) were other, and 38 (13.4%) preferred not to answer (Table 1). Of the respondents, 22.7% have been in practice for 5 years or less, while 30.7% have been in practice for over twenty years. In regards to location, the distribution of United States-based respondents can be found in Figure 2. A total of 36.6% of the responses came from those practicing in Massachusetts, and other popular locations included California (9.8%), New York (7.2%), and Texas (5.1%). Nine respondents (3.2%) answered “other” when asked about their location. A majority of the survey responses (59.7%) were from physicians (both MDs and DOs), but other healthcare professionals were also represented in the sample, including registered nurses (12.9%), nurse practitioners (4.2%), and physician’s assistants (1.4%). A differential analysis was performed to test if there was a difference in responders to survey questions based on specialty. This analysis was conducted by comparing the three most represented specialties by frequency to the whole. No significant difference was found amongst any of the groups.

### 3.2. Distribution of Specialty

A diverse group of specialties was accounted for within the survey responses. Most notable were the specialties of pulmonology (23.3%), cardiology (16.7%), and family or internal medicine (13.2%). A full list of the specialties represented is included in Table 2. In our 2019 survey, we found that there was a significant difference in the way that cardiologists perceived POCT compared to other specialties [7]. We ran a similar analysis for this year’s survey and observed no significant difference in the way that any of the three most common specialties answered (pulmonology, cardiology, family or internal medicine) when compared to the other respondents as a whole.

### 3.3. Important Aspects of POCT

Participants were asked to select the first, second, and third most important characteristics of point-of-care technology when incorporating it into their regular practice. Accuracy was the most selected response, with 239 (82.3%) respondents selecting to put accuracy in their top three choices. Other responses with high selection by respondents included ease of use (189, 65.9%), availability of testing (104, 36.2%), cost of testing (93, 32.4%), and that it does not disrupt workflow (84, 29.3%). (Table 3).

### 3.4. Benefits of POCTs 

Participants were presented a series of 15 statements regarding the benefits of POCT and asked to rate the degree to which they agreed with these beneficial statements (Figure 3). For all items, a majority of respondents agreed/strongly agreed with the statement. The statement “POCTs improve patient management” was the most agreed-with, with 92% of participants selecting agree or strongly agree in the survey. The second most agreed-with statement was “POCTs improve clinician confidence in decision making,” with 87.5% of respondents selecting agree or strongly agree. The third most agreed-with statement was “POCTs enable more effective targeted treatment” (83.6%).

### 3.5. Concerns over POCTs

Participants were presented with a series of 14 statements regarding possible concerns over POCT, and were asked to rate the degree to which they agreed with these concerns (Figure 4). For all of these choices, a minority of respondents agreed or strongly agreed with the statement. In fact, for a number of statements, only a small percentage of respondents agreed: six statements had less than 10% agreement, and another four had 10.1–19.9% agreement. The statement “POCTs lead to over-testing” was the most agreed-with statement, with 30% of participants selecting agree or strongly agree in the survey. The second most agreed-with statement was “I might not be reimbursed for the cost of POCT,” with 29.3% of respondents selecting agree or strongly agree. 

Some of these common statements of concern were further divided into statements regarding accuracy and use, and statements regarding finances. The statements regarding accuracy and use were “Diagnostic accuracy of POCTs is not good enough to make a clinical decision,” “POCTs are too difficult to use,” and “The results of POCTs are difficult to interpret/not definitive.” The statement “Diagnostic accuracy of POCTs is not good enough to make a clinical decision” was the most agreed-with statement regarding accuracy and use, with 16.4% of participants selecting agree or strongly agree. The statements regarding finances were “Equipment costs associated with POCTs are too high,” “Staff training costs associated with POCTs are too high,” and “I might not be reimbursed for the cost of the POCT.” The statement “I might not be reimbursed for the cost of the POCT” was the most agreed-with statement regarding finances, with 29.3% of participants selecting agree or strongly agree in the survey. 

### 3.6. Product Adoption Practices

Participants were given six statements regarding the adoption of POCT technology in their medical practice (Figure 5). The statement “adoption of new lines of products or service is often constrained by available resources” was the most agreed-with statement regarding adoption, with 55.7% of respondents indicating that they agree or strongly agree. The statement “my practice typically adopts a cautious wait-and-see posture in order to minimize the possibility of making costly decisions” was the second most agreed-with statement, with 40.4% of respondents indicating agree or strongly agree (Figure 5). 

### 3.7. COVID-19 Questions

Healthcare providers also answered several questions on the use of POCT for the COVID-19 pandemic (Table 4). A total of 64.5% of responses agree or strongly agree that POCT is improving patient care during the COVID-19 pandemic. Most responses agree that POCT improves the diagnosis of patients with COVID-19 (56.5%), and agree that POCT has increased community access to COVID-19 testing (59.9%). A slight minority (46.0%) of respondents agreed that POCT has been beneficial in decreasing transmission of COVID-19. When asked about their experience with POCT during the COVID-19 pandemic, 44.6% said they had an excellent/very good/good experience, 42.9% said they had no experience, and 11.2% reported a poor or very poor experience. When asked to categorize their level of experience with POCT during the pandemic, the most common responses were moderate (37.3%) and minimal (34.5%). 

### 3.8. POCT Glucose Testing Changes over Time

Increased use of POCT over time may contribute to the generally positive responses observed in this study. To verify that POCT use was in fact increasing, we conducted a retrospective review of the use of hospital bedside testing for glucose management, a common inpatient test at our hospital. There were 333,266 glucose POCTs performed at our 600-bed academic medical center during 2018, 346,540 during 2019, 337,644 during 2020, and 364, 968 during 2021. The lower number of tests in 2020 was associated with the modification of glycemic monitoring frequency for patients with COVID. These overall usage measures correspond to a 9.8% increase from 1.52 POCT glucose tests/licensed bed/day in 2018 to 1.67 in 2021. 

## 4. Discussion

In this survey of healthcare professionals, we found that most respondents perceived an unmet need for POCTs that diagnose or manage/monitor patient care. A majority of participants agreed with statements related to the benefits to POCT, while a minority agreed with statements describing possible concerns over POCT. Interestingly, a larger proportion of respondents agreed or strongly agreed with common concerns regarding finances associated with POCT than agreed or strongly agreed with common concerns regarding accuracy and use. 

The relatively large concern regarding the finances of POCT is likely a function of many different factors surrounding the change in workflow brought on by POCT, and the cost/reimbursement of the services that will be provided. For example, a multitude of consumer technologies (e.g., Apple Watch, KardiaMobile, etc.) capable of accurate cardiac monitoring are currently available and used without the advice of a medical professional, inverting the traditional dynamics of provider-initiated patient care to a technology-initiated patient care [13]. With patients now generating and obtaining their own healthcare data without the assistance of their physician, they are more empowered to understand their health; however, it is unclear how patients will use these data to interact with their physician, and how physicians will be reimbursed for this care [13]. For example, before a patient with chronic hypertension leaves their bi-annual exam, they will typically make their next appointment about 6 months from that date. In this scenario, the patient knows that their hypertension will be examined by an expert in 6 months, and the physician knows that they will be reimbursed in 6 months for this next visit. As we shift to a model where technology initiates care, it is not clear what will trigger patients to go and seek help, or insurance companies to reimburse physicians. Physicians do not know if they will get reimbursed for visits scheduled based on data from wearable devices, if their standard appointment intervals will remain despite the influx of data, or if insurance companies will decide to set thresholds in which certain data must be collected before the next reimbursable visit. In many cases, physicians are unclear about how to navigate the shift away from provider-initiated patient care, and unsure about the billing changes that will come with this shifting paradigm. 

Additionally, in a technology-initiated workflow, physicians may have to be up to date with various POCT devices and the cost associated with those devices to ensure quality care for their patients. The severity of any potentially inaccurate or misdiagnoses could be detrimental to patients’ health, and significantly affect the level of care physicians provide at each visit. If, for example, a patient can only afford a lower quality POCT device, a physician may struggle to interpret or trust the data produced by this device. It is not clear whether physicians will bear the cost of educating themselves about the various technologies, nor how insurance will reimburse physicians for their interpretation of the data produced by varying devices. Additionally, it remains to be seen whether the patient, physician, insurance companies, or some joint decision process will be used when selecting a POCT, and what factors—such as data accuracy, validation, cost, and security—are most important when making a decision.

## 5. Conclusions

The past decade has ushered extensive digitization into the US healthcare system, with significant advances in POCTs [4]. As new point-of-care technologies emerge, healthcare providers are enthusiastic about the potential benefits of POCT on both patient management and clinician decision making. However, these same healthcare providers have many valid concerns, including the over-testing of their patients and the financial cost of these devices to their practice. To harness this enthusiasm as well as address these concerns, there must be continued research into the tangible cost of POCT adoption, and greater collaboration among industry partners, healthcare professionals, and patient groups.

## Figures and Tables

**Figure 1 diagnostics-12-00533-f001:**
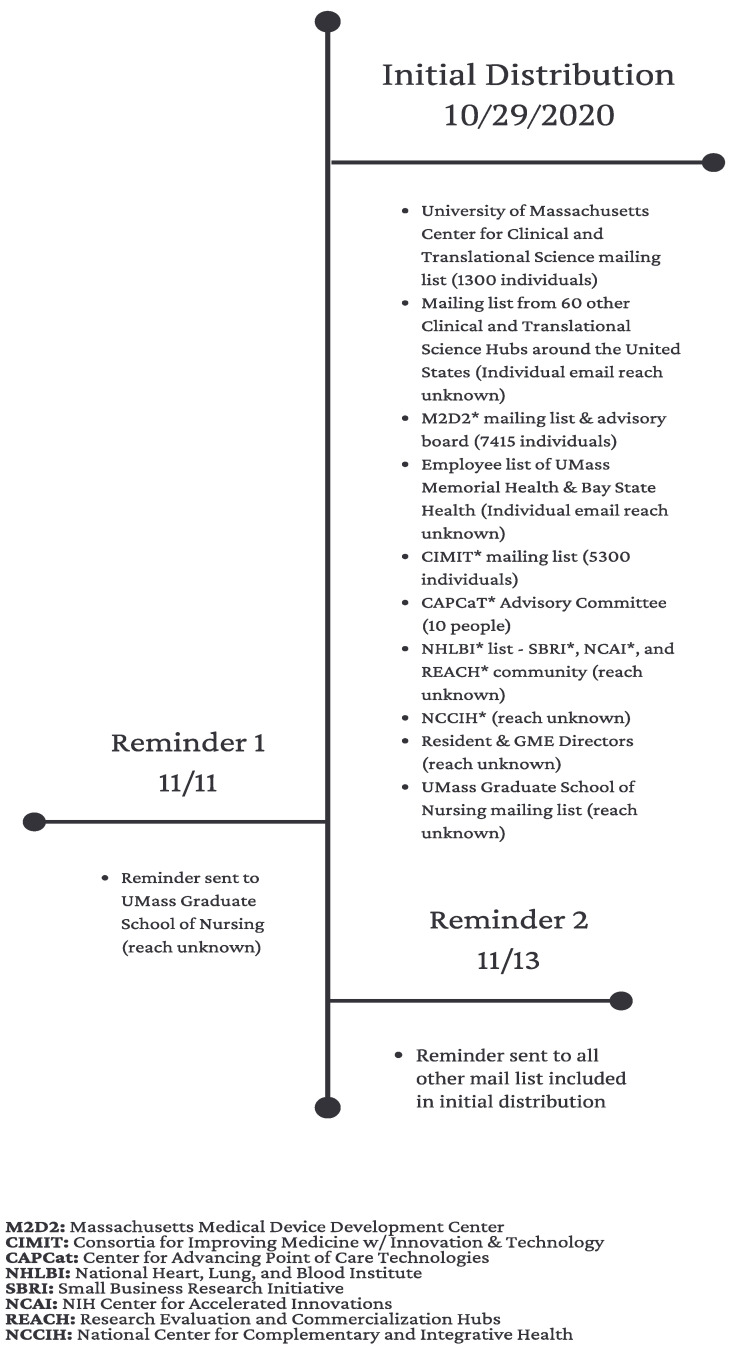
Timeline of survey distribution via various mailing lists.

**Figure 2 diagnostics-12-00533-f002:**
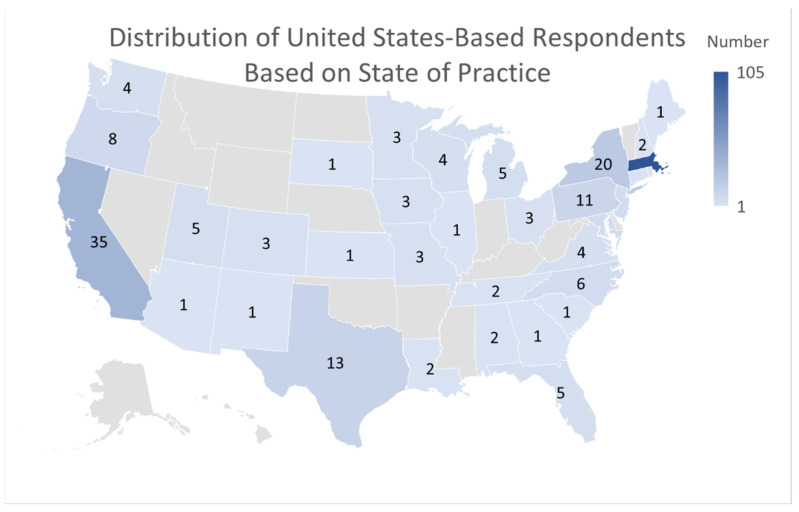
Distribution of United States-Based Respondents Based on State of Practice. Powered by Bing © GeoNames, Microsoft, TomTom.

**Figure 3 diagnostics-12-00533-f003:**
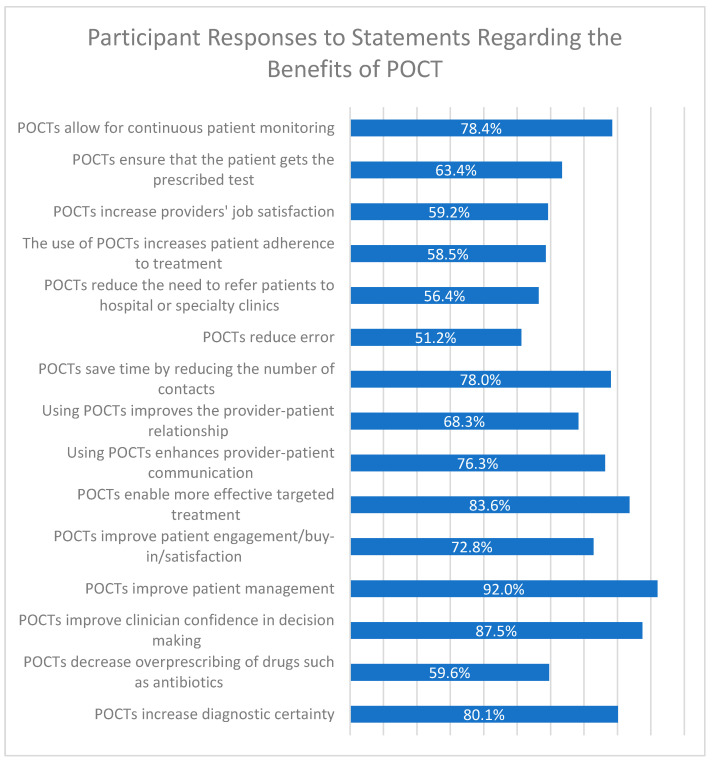
Participant responses to statements regarding the benefits of POCT. Survey respondents were given 15 statements regarding the potential benefits of POCT, and were asked to respond that they “strongly agreed”, “agreed”, were “neutral/not sure”, “disagreed”, or “strongly disagreed” with the statement. The percentage of respondents who said they agree or strongly agree is shown to the right of each statement.

**Figure 4 diagnostics-12-00533-f004:**
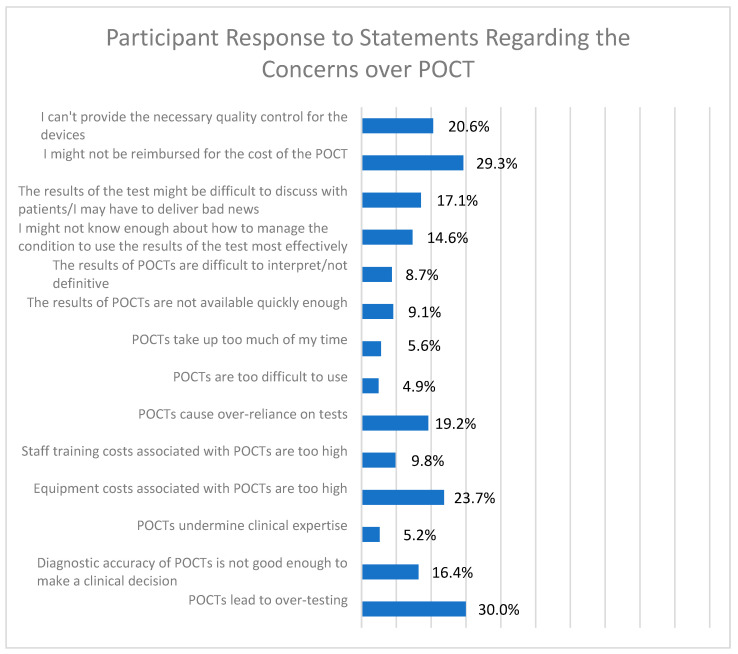
Participant response to statements regarding the concerns over POCT. Survey respondents were given 14 statements regarding potential concerns over POCT and were asked to respond that they “strongly agreed”, “agreed”, were “neutral/not sure”, “disagreed”, or “strongly disagreed” with the statement. The percentage of respondents who said they agree or strongly agree is shown to the right of each statement.

**Figure 5 diagnostics-12-00533-f005:**
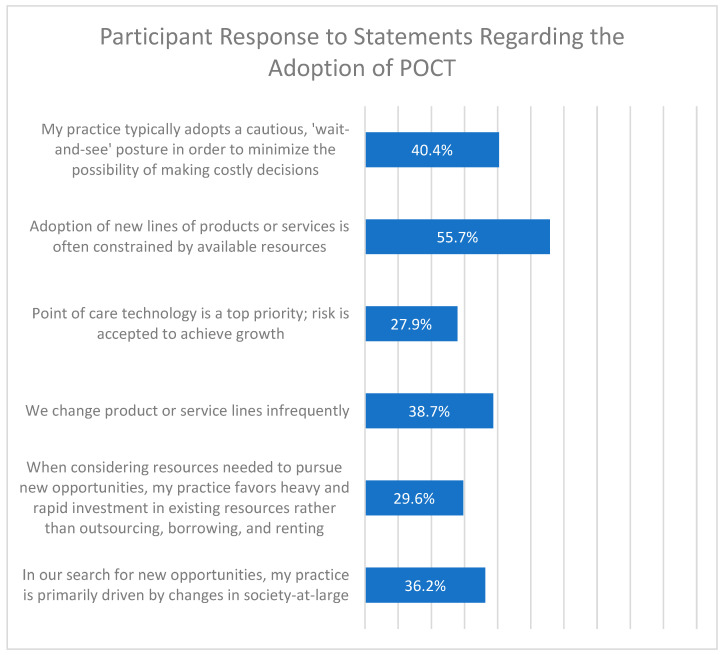
Participant response to statements regarding the adoption of POCT. Survey respondents were given 6 statements regarding adoption of POCT in their clinical practice and were asked to respond that they “strongly agreed”, “agreed”, were “neutral/not sure”, “disagreed’, or “strongly disagreed” with the statement. The percentage of respondents who said they agree or strongly agree is shown to the right of each statement.

**Table 1 diagnostics-12-00533-t001:** Demographics of survey respondents.

Participant Demographics (*n* = 287)	Number of Respondents (%)
Gender	
Male	154 (53.7)
Female	120 (41.8)
Undisclosed	13 (4.5)
Race	
White	191 (66.6)
Black or African American	9 (3.2)
Asian	47 (16.5)
American Indian or Native Alaskan	1 (0.4)
Other	4 (1.4%)
Preferred Not to Answer	38 (13.4%)
Years in practice	
0–5 years	65 (22.7)
6–10 years	36 (12.5)
11–15 years	51 (17.8)
16–20 years	37 (12.9)
Over 20 years	88 (30.7)
Undisclosed	10 (3.5)
Profession	
Physician (MD/DO)	171 (59.6)
Advanced Practice Providers (NP/APN/PA)	20 (7.0)
RN-Registered Nurse	37 (12.9)
Other	52 (18.1)
Undisclosed	7 (2.4)
Patient Practice Environment	
In-hospital	151 (52.6)
Ambulatory Care	72 (25.1)
ER	16 (5.6)
In-home	8 (2.8)
Other	29 (10.1)
Undisclosed	11 (3.8)

**Table 2 diagnostics-12-00533-t002:** Healthcare professional respondents by specialty.

Specialty	Number of Respondents (% of Respondents)
Cardiology	48 (16.7)
Family or Internal Medicine	38 (13.2)
Pulmonology	67 (23.3)
Hematology	4 (1.4)
Emergency Medicine	19 (6.6)
Sleep Medicine	4 (1.4)
Other	101 (35.2)

**Table 3 diagnostics-12-00533-t003:** Most Important characteristics of a point-of-care technology.

Characteristic	Number of Times Listed in Top 3 (% of Respondents)
Accuracy	239 (82.3)
Ease of use	189 (65.9)
Availability	104 (36.2)
Cost of Testing	93 (32.4)
Does Not Disrupt Workflow	84 (29.3)

**Table 4 diagnostics-12-00533-t004:** Participant response to statements regarding COVID-19. (top) Survey respondents were given 4 statements regarding the COVID-19 pandemic and its impact on POCT in their clinical practice, and were asked to respond that they “strongly agreed”, “agreed”, were “neutral/not sure”, “disagreed”, or “strongly disagreed” with the statement. (bottom) Survey respondents were asked what their experience with POCT was like during the COVID-19 pandemic and were then asked how much experience they had with POCT during the COVID-19 pandemic. The number of respondents (percentage) who said they agree or strongly agree is shown to the right of each statement.

Statement	Number Agree/Strongly Agree (%)
POC testing use is improving patient care during the COVID-19 pandemic	185 (64.5)
POC testing improves diagnosis of patients with COVID-19	162 (56.5)
POC testing has been beneficial in decreasing transmission of COVID-19	132 (46.0)
POCT has increased community access to COVID-19 testing	172 (59.9)
What is your experience with POCT during the COVID-19 pandemic?	
excellent/very good	19 (6.6)
good	109 (38.0)
not sure/no experience	123 (42.9)
poor	22 (7.7)
very poor	10 (3.5)
What is your amount of experience with POCT during the COVID-19 pandemic?	
substantial/a lot	34 (11.9)
moderate	107 (37.3)
minimal	99 (34.5)
none	43 (15.0)

## Data Availability

Please contact the authors for a copy of the complete data set.

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
