# Peer review of "Understanding Healthcare Professionals’ Perspectives on Point-of-Care Testing"

_diagnostics, 2022, doi:10.3390/diagnostics12020533_

Round 1

Reviewer 1 Report

This manuscript describes the perspectives to POC of healthcare professionals.

The topic is met with Journal’s scope.

Study design;

  • Authors invited potential responders via email. How did authors find or select potential responders. Please clarify include and exclude criteria and process.
  • Please indicate sample size calculation if you calculated. If not, please describe reason why you did not.
  • Did you validate the questionnaire? Please indicate the detail of validation process of questionnaire.

Results and discussion:

  • Participants have a variety of professional backgrounds. The perspective to POC is considered to different depending on their profession or specialty. I recommend to analyzing and discuss the variation of professional background on perspectives to POC. It would be useful for most readers.

Reviewer 2 Report

The manuscript “Understanding Healthcare Professionals Perspective on Point-of-Care Testing” by Sean Teebagy et.al., summarizes a detailed survey study to understand the current perspective of healthcare professionals towards POCT. The concept and idea of the work is good, but I would request the authors to revisit their manuscript and readdress all their data collected as tables and make visually appealing and easy to understand graphs and  figures. A lot has been presented, but the main idea of it is missing. Readers do not have time to spend hours in a manuscript. And in such cases, the manuscript loses its scientific point, despite of the work being good like yours. I will proceed with a decision once changes are addressed. 

However, Kindly revise as below:

  • ‘……….October 2020 and November 2020. Why the survey is done over a very short period of time?
  • I would request the authors to kindly read the Journal guidelines before submitting a manuscript. There are Line number missing, in order to track the manuscript.
  • “While prior work has documented the advantages of POCT, little is known about the adoption of POCTs by healthcare providers”. Was this include in the survey/questionnaire study?
  • Kindly present the survey strategy pictorially, steps by steps or as a flowchart to help understand the concept of this study. For example, Study Goal, Participant selection, Gender, age, Strategy of execution , email sent, second email sent, data collected, results and discussions
  • Most questions were assessed on a 5-point Likert-like scales, ranging from “strongly disagree” to “strongly agree”. Kindly present this as a Figure x., with a

Question x: ………?

Answer x:  a____, b_____, c______, d_____

Score :

  • Section 3.1. Demographics & Table 1, Kindly present this a graph or a colored Venn diagram.
  • Section 3.1. “In regard to location……”, Kindly include a world map and highlight areas from where this was collected. Kindly understand that collecting good data does not “qualify” for an acceptable paper/article. Drafting the manuscript in a readable way is very important for a good paper. Kindly put some time to re-address all the data in a presentable manner rather than just throwing in a bunch of tables which cannot be understood at all. This article has a lot of good data, that needs to be addressed and presented in a readable format.
  • Section 3. Results. Kindly address these few points before discussion of each section or analysis or tests? Each of the below listed questions should be the way to address any result or discussion. This way the readers can understand the propose or motto of doing that study rather than presenting your results.
    1. Why demographic, distribution study?
    2. What is the purpose or goal?
    3. What information does demographic, distribution analysis can give about the POCT?
    4. What statistical method is chosen to evaluate?

Kindly re-address each result/test section with the above questions. 

  • Kindly elaborate more on the introduction sections, and the result and discussion sections with references. Appropriate references are required to support the claim and results of any study.

Round 2

Reviewer 1 Report

The manuscript was revised very well. I don't have additional comments.

Reviewer 2 Report

Kindly accept in present format.